# Seeing is Knowing: Advancing Semantic Understanding with MLLMs in Grounding Tasks

## Abstract

Large vision models (VLMs) have achieved significant success in most daily scenarios but face challenges in specialized grounding tasks. This limitation is primarily due to insufficient semantic understanding of both tasks and images in current vision models. In contrast, large multimodal language models (MLLMs) excel in semantic comprehension and instruction-following but underperform in detailed recognition. To harness the strengths of both, we propose utilizing MLLMs to assist VLMs in handling difficult segmentation tasks. Our approach involves: (1) leveraging MLLMs for their semantic expertise, and (2) design effective framework for zero-shot segmentation. Our proposed framework is generalizable and performs well across various tasks. Experimental results show a significant performance improvement (10%+) in challenging tasks such as camouflage object detection, anomaly detection, and medical image segmentation compared to zero-shot baselines.

## 1 Introduction

Zero-shot challenging segmentation has become an essential area of research due to the its significant applications in fields such as medical imaging and anomaly detection Trinh (2023); Cao et al. (2023b). Traditional segmentation methods struggle with recognizing objects in challenging zero-shot scenarios Chen et al. (2023a); Tang et al. (2023), where data scarcity and lack of fine-grained labels pose substantial challenges.

We find that the primary challenges for Large Vision Models (LVMs) in such task is not the visual recognition ability. As shown in (a) in Figure 1, Table 1 and Table 3, the LVMs are able to accurately segment targets within a few visual prompt and even under 10% errors. However, when it is asked to auto-segment the target, it almost fails completely. Therefore, we conduct experiment in this paper and identify the bottleneck is the semantic knowledge.

On the contrary, we find that Large Multimodal Models are good at semantic understanding, from Figure 1 and Table 3. Therefore, our question is: Can we transfer the semantic expertise from MLLMs to LVMs for challenging zero-shot segmentation? There have been previous works on it, but they are either (1) directly grounding the target with grounding-capable MLLMs Zhang et al. (2023b), which suffers from low accuracy (2) Using weak labels to train an Adapter Chen et al. (2023a), which requires training (3) Introducing extra smaller models for guidance Tang et al. (2023) which hurts generalization ability, (4) Segmenting-then-picking by MLLMs, which does not fall in the scope of challenging segmentation since segmenting is already difficult for LVMs. To overcome the limit, we propose *De*composed *S*egmentation with *Se*lective *R*e-localizing(*DeSSeR*), a training-free, zero-shot while highly generalized method. As shown in FIgure 1 (b), our method use decomposition to approach the question in a coarse-to-fine grained manner, and fully utilize the semantic ability of powerful MLLMs to correct unreliable results. Abundant analysis is done to illurstrate the design of the method, and experimental results on three dataset, Camouflage Object Detection (COD), Zero-Shot Anomaly Detection (ZSAD), and polyp medical image segmentation (polyp) demonstrate the superiorness of our method.

In conclusion, our contributions are:

1. Demonstrating that zero-shot challenging segmentation benefits significantly from the semantic expertise of MLLMs.

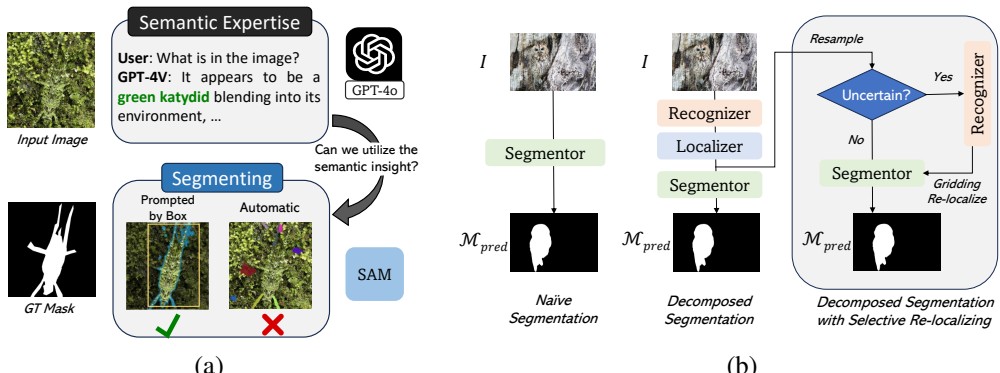

Figure 1: (a) Current limitations of LVMs in challenging zero-shot segmentation is the lack of semantic understanding while LMMs is able to semantic expertise. (b) Comparison of our framework with naive segmentations.

2. Quantitatively showing that decomposing segmentation tasks improves performance in challenging zero-shot scenarios.

3. Identifying the limitations of grounding models in localization and proposing uncertainty estimation methods, as well as gridding localization techniques that utilize the generalization ability of MLLMs.

4. Composing a novel strategy and introducing DESSER, a training-free framework. Experimental results on camouflage object detection, anomaly detection, and polyp segmentation show significant performance improvements (over 10%) compared to zero-shot baselines, achieving results comparable to fully-supervised methods.

## 2 PRELIMINARIES

### 2.1 PROBLEM SETTING

In challenging zero-shot segmentation, a model $f : I, p \rightarrow M$ is given an image $I$ and outputs a prediction mask $M$ based on the task $t$ and prompt $p$. Unlike traditional segmentation, the model is not trained on the specific domain, making it desirable for computational convenience and data scarcity.

To tackle the 0-shot, we utilizing generalizing ability from large pretrained models OpenAI (2023a); Caron et al. (2021), including vision and large multimodal models. Vision models is transformer-based, and MLLMs are autoregressive, generative language models conditioned on the visual modality. The prompt $p$ is designed to guide the MLLMs to do the task. The answer sampled from $p(a|v, q)$ is used to enhance the segmentation of vision models. The setting is common in previous works, Tang et al. (2023); Biswas (2023); Zhang et al. (2023b). Our method differs in that it can generalize to multiple tasks, all with large performance gain, and just require necessary changes to prompt.

### 2.2 MOTIVATING EXPERIMENT

**Is semantic expertise helpful in zero-shot segmentation?** Per previous work that distill guidance from MLLM and use LVM to segment, they either Recognizer/Localizer/Segmentor. Experiment are conducted to see if that works generally and in zero-shot. we conduct naive decomposition strategy to test if semantic expertise is beneficial for segmentation. To select the data, we randomly picked 100 images from CAMO dataset (in camouflage object detection filed), and simply use MAE to test the result. For models, we use gpt-4o to generate semantic expertise, and SEEM for segmentation. According to Table 3 and Figure 2, we compare two ways: (1) Asking SEEM to segment directly (2) Asking SEEM to segment based on the name of the object. Results show when the original MAE is large, meaning SEEM may totally not know the target, the improvement from semantic expertise is huge. This demonstrates the lack of semantics is the main reason for failed segmentation.

**How to exploit semantics effectively?** Is the semantics being exploited by the naive semantic-enhanced text prompt? From Table 3, we find that recognizing from MLLMs is high, there are still complete misses; when given rough grounding (boxes with 10% off accuracy), the segmenting is way higher. Therefore, we claim that the segmentor model is good at segmenting, but lacks the ability in between, which we call it "localization". So when we want it to localize and segment together, it suffers from the localizing. To further prove the claim, we add a localizer MLLM in between, composing a three-step pipeline as shown in Figure 1. It shows that the rough localization by MLLM helps transferring semantic knowledge, for that it can (1) understand better semantics, according to high IOU of box than vanilla SEEM, (2) is able to better prompt SEEM, visual prompt.

In conclusion, we study the challenging zero-shot segmentation and propose to decompose it in three parts, recognizing, localization (rough grounding) and segmentation (exact grouonding). We want to see how to optimize each and their connect to improve the performance.

## 3 ENHANCING SEMANTICS THROUGH DECOMPOSITION

As suggested in Section 2.2, we can use decomposition to advance semantic expertise from MLLM in segmentation models. In this section, we put our main focus on the decomposition of localization and segmentation. Recognizing alone is not studied in detail since (1)this can be seen as a VQA question, and plenty of them have been discussed Khan et al. (2024) (2) We focus more on the correlation between visual question and visual answer, Section 3.5. For localization, we mainly study how to transfer semantic from recognizer to better localization result. For segmentation, since the whole process is end to end, we mainly study how to provide effective visual prompt for LVMs.

In the section, all experiments are conducted based on CAMO.

### 3.1 SHOULD LOCALIZATION BE DECOMPOSED?

We examine if the decomposition of localizing is useful. As shown in the Table 3, we test naive end-to-end localizing with prompt:*"Where is the camouflage object?"*[1] ,and simple decomposition for localization, recognizing name of the object first then localize: Two prompts are used in order. (1)prompt:"What is the name of camouflage object?", (2)prompt:"Where is the [name]?". Multiple MLLMs (both close-source and open-source are tested) are tested, and results show in localization, they all benefit from such decomposition. This is consistent with what previous similar work proves in VQA and language field Zhou et al. (2022); Khan et al. (2024): If we ask model to focus on one thing at a time, it generally produces better result.

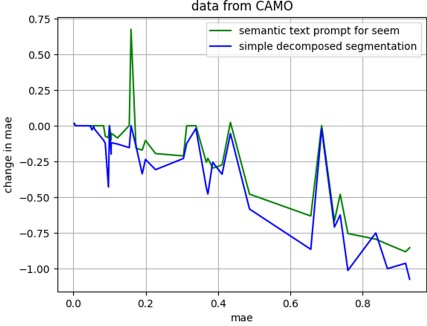

Figure 2: Semantic expertise and decomposition helps.

| Model | Visual Prompt | MAE |
|-------|---------------|-----|
| SAM | auto-segment | 0.412 |
| | exact bbox | **0.057** |
| | random-10%-off bbox | 0.083 |
| | random-30%-off bbox | 0.090 |
| | random-50%-off bbox | 0.138 |

Table 1: SAM segment images from CAMO with different visual prompts. It can be found that rough localization is able to improve segmentation largely.

---

[1] one thing to note is that the model sometimes do decomposition themselves, but not much impact on the result

| Method | LVLM-based | Zero-Shot | Training Settings | CAMO | | | CVC300 | | | VisA* | |
|---|---|---|---|---|---|---|---|---|---|---|---|
| | | | | $S^\alpha\uparrow$ | $F_\beta^a\uparrow$ | MAE↓ | mIOU↑ | mDice↑ | MAE↓ | $AUROC_{px}\uparrow$ | $F1\text{-max}_{px}\uparrow$ |
| **Task-Specific Methods** | | | | | | | | | | | |
| diffCOD Chen et al. (2023b) | ✗ | ✗ | F | 0.795 | 0.704 | 0.082 | - | - | - | - | - |
| ZoomNeXT(ResNet) Pang et al. (2023) | ✗ | ✗ | F | 0.833 | 0.774 | 0.065 | - | - | - | - | - |
| ZoomNeXT(PVT) Pang et al. (2023) | ✗ | ✗ | F | **0.888** | **0.859** | **0.040** | - | - | - | - | - |
| WSCOD He et al. (2023) | ✗ | ✗ | W | 0.735 | 0.641 | 0.092 | - | - | - | - | - |
| CVPLF Tang et al. (2023) | ✓ | ✓ | N | 0.700 | 0.650 | 0.100 | - | - | - | - | - |
| PraNet Fan et al. (2020b) | ✗ | ✗ | F | - | - | - | 0.797 | 0.871 | 0.010 | - | - |
| Polyp-PVT Dong et al. (2021) | ✗ | ✗ | F | - | - | - | 0.833 | 0.890 | 0.007 | - | - |
| Meta-Polyp Trinh (2023) | ✗ | ✗ | F | - | - | - | **0.862** | **0.926** | 0.006 | - | - |
| SAM-CLNet Zhao et al. (2023) | ✗ | ✗ | F | - | - | - | 0.800 | 0.876 | 0.008 | - | - |
| Polyp-SAM++ Biswas (2023) | ✓ | ✓ | N | - | - | - | 0.690 | 0.730 | - | - | - |
| GPT-4V-AD Zhang et al. (2023b) | ✗ | ✓ | N | - | - | - | - | - | - | 77.9 | 9.3 |
| WinCLIP Jeong et al. (2023) | ✗ | ✓ | N | - | - | - | - | - | - | 81.2 | 15.8 |
| SAA Cao et al. (2023a) | ✗ | ✓ | N | - | - | - | - | - | - | 84.7 | 13.2 |
| SAA++ Cao et al. (2023a) | ✗ | ✓ | N | - | - | - | - | - | - | 77.8 | 29.8 |
| **General Vision Models** | | | | | | | | | | | |
| OpenSeeD Zhang et al. (2023a) | ✗ | ✓ | N | 0.592 | 0.552 | 0.201 | 0.301 | 0.402 | 0.060 | 60.2 | 2.2 |
| SAM Kirillov et al. (2023) | ✗ | ✓ | N | 0.684 | 0.606 | 0.132 | 0.685 | 0.742 | 0.016 | 69.8 | 7.5 |
| SAM-HQ Ke et al. (2023) | ✗ | ✓ | N | 0.701 | 0.625 | 0.114 | 0.679 | 0.722 | 0.014 | 68.3 | 5.5 |
| SEEM Zou et al. (2023) | ✗ | ✓ | N | 0.697 | 0.578 | 0.121 | 0.674 | 0.696 | 0.020 | 71.1 | 6.3 |
| **LVLMs with Segmentation Ability** | | | | | | | | | | | |
| LISA Lai et al. (2023) | ✓ | ✓ | N | 0.690 | 0.625 | 0.148 | 0.152 | 0.221 | 0.059 | 63.1 | 1.2 |
| Ours | ✓ | ✓ | N | 0.833 | 0.818 | 0.064 | 0.845 | 0.903 | **0.004** | **86.4** | **32.5** |

Table 2: **Quantitative results overview.** The table organizes compared methods into three categories, arranged from top to bottom: Task-Specific Methods (features numerous '-' entries, indicating their limited applicability across different datasets), General Vision Models, and LVLMs with segmentation ability. 'VisA*' specifically refers to using only anomaly images from the VisA dataset for segmentation-focused evaluations. Training settings are indicated as 'N' (no supervision), 'F' (full supervision), and 'W' (weak supervision). **Bold** and underline highlight the best and second-best results, respectively. The results showcase our method's competitive edge against fully-supervised approaches and substantial improvements over weakly or unsupervised methods across all datasets.

## 3.2 HOW WELL CAN DECOMPOSED LOCALIZATION BE?

Can the simple decomposition be further improved? First, previous works show that customized prompt for each task help improve the result of MLLMs Zhang et al. (2023b); Zhou et al. (2023); Tang et al. (2023). However, that is not in the scope of this paper and we try to focus on the general aspect. There are two problems identified.

**Grounding Multiple Instances.** MLLMs are hard to ground multiple objects if not explicitly prompted. Recent works Rasheed et al. (2023); Wu et al. (2023) also reveals this. For example, when asked to localize "people", localizer MLLMs inclined to localize either one person or the whole people, but not every one in one box individually. This compromise the result of segmentation. To overcome the limit, we include instance-aware-prompt: that is we generate one query for each instance target, and then we prompt localizer to answer grounding for each queries. Experiment results can be shown in Table 6 to suggest its effectiveness.

**Target Misses.** Though grounding MLLMs are good at understanding semantic, there are cases they still miss the recognized targets or only able to identify part of it. We attribute this to two reasons, as shown in the left image of Figure 3: 1. referring text confusion (text prompt): The description for the target is not clear enough. As shown in the upper row, the prompt is misleading since the localizer can not know what is "bend" without refering to normal objects. Even though in this case we can exactly use "the rightmost metal legs" to guide the localizer, the way to speicify such prompt varies largely across images, and it is hard to ensure the exact expression all the time. 2. The localizer fails to understand the image,, even with the guidance of semantic expertise. As shown in the lower row, The localizing ability of a model is generally weaker than the semantic understanding ability for much less grounding data trained than image-text pairs data.

Therefore, we wonder (1) can we identify problematic localization? (2) if there is a way to further exploit the semantics for localization from recognizer?

## 3.3 SELECTIVE RE-LOCALIZING WITH MLLM GRIDDING

To address the above issues, we propose the strategy: selective re-localizing with MLLM Gridding. It can first estimate the uncertainty of MLLMs, then use recognizer MLLM to improve the localiztion.

### 3.3.1 UNCERTAINTY ESTIMATION

No matter is the confusing referring text prompt or the inability, the localizer is not sure about its predicctionl. Therefore, we propose to use sampling for uncertainty estimation. We test **consistency over multiple answers**, and select those with inconsistent answers.

After selection, we combine the answers in one image and ask recognizer to confirm which one is correct.

|  | Model | Text Prompt | Metrics | Result |
|---|---|---|---|---|
| Recognizing | GPT-4o | What is the name of camouflage object? | Accuracy | **0.88** |
|  | QWen-VL-Max | What is the name of camouflage object? | Accuracy | 0.80 |
| Localization | Qwen-VL-Max | What is the name of camouflage object? | IOU | 0.72 |
|  | Qwen-VL-Max | What is the [object name]? | IOU | **0.81** |
|  | CogVLM-14B | What is the name of camouflage object? | IOU | 0.65 |
|  | CogVLM-14B | What is the [object name]? | IOU | **0.81** |
|  | SEEM (box of pred.) | camouflage object | IOU | 0.46 |
|  | SEEM (box of pred.) | [object name] | IOU | 0.66 |
| Segmentation | SAM | Bbox | MAE | 0.057 |
|  | SAM | Bbox, center point | MAE | 0.048 |
|  | SAM | Bbox, center point, 1 rand. point | MAE | 0.041 |
|  | SAM | Bbox, center point, 3 rand. point | MAE | **0.039** |
|  | SAM | Bbox, center point, 5 rand. point | MAE | **0.039** |
|  | SAM | Center point | MAE | 0.061 |
|  | SEEM | Center point | MAE | 0.063 |

Table 3: Comparison of prompts for recognizing, localizing and segmenting camouflage objects on subset data from CAMO. "[object name]" refers to the category name of the object in the image. "SEEM (box of pred.)" means that we use the bounding box of its prediction to calculate IOU. "Center point" refer to the center of the bounding box. "rand. point" refer to random sampled points within the bounding box.

**Recognizer's verification ability**: One of MLLM's emerging ability is verification on visual prompt, as show in Figure 3. The recognize is able to select the grid box that cantains the target. This prompt us to provide a novel localization strategy when loicalizer failed.

### 3.3.2 MLLM GRIDDING LOCALIZER

Therefore, we propose gridding prompt for recognizer. As shown in Figure 3, the gridding is laid on the image and sent to MLLM. Then the overall gridding is cropped based on the whole bounding box, and this is sent to localization for finer localize.

**Replacing localization with gridding?** Totally replace localization with MLLM localizer hurts overall performance, because when localizer know the object, its localizing accuracy is much higher than MLLM recognizer. Also, it is computational consuming.

### 3.4 SEGMENT WITH LOCALIZATION RESULT

In this subsection, we dicuss how to segment based on localization result. More specifically, we use SAM to segment, which takes pure visual prompt. We study what visual prompt is valuable, and how to achieve them.

**Various visual prompt for segmentation** We conduct experiment on different visual prompt for segmentation, the results show that box prompt is the most effective one. Also, we find that providing points used for visual prompt is also important, even though a fixed point prompt may help SAM's segmenting. Several visual prompting is studies including (1)Single point (2) Box (3) Center Point in the Box (4) Corner Point of the box, as shown in 3, 5.

**Cost-effective visual prompt** The localizer only provide box; to utilize the point prompt, we can use recognizer. According to Table 5, we find that even the fixed point prompt with box performs

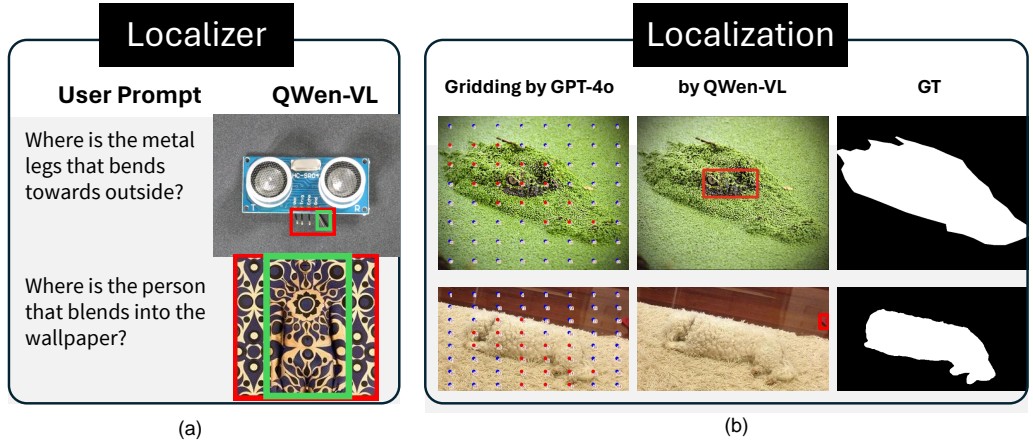

Figure 3: (a) Cases where localization fails. The above row demonstrate cases where prompt is confusing (not knowing what is counted as "bending outside". Below row shows localizer failed to understand images. "red box" is the predicted grounding while "green" is the GT. (b) Different ways to do localization. Here, we show cases where Localizer (QWen-VL) failed while gridding localizing by recognizer (GPT-4o) succeds. The point represent the center of the grids, red for positive and blue for negative.

quite closely with recognizer-identified point prompt. Therefore, we adoopt fixed point prompt with box promtp as a cost-effective way.

## 3.5 SCOPE AND LIMITATIONS

Our method focus specifically on the intersection of VQA and grounding for segmentation, so not including pure text output such as optimizing recognition stage. Our method focus on zero-shot segmentation, all the models are not trained. Currently, the performance of un-trained models are still far from trained domain experts, but the method open a new perspective in regards to segment by utilizing the generalizing power of MLLMs. We choose to enable our methods on black-box models, for wider application. We do not target on only one task (focus on one optimize), but to find what can be generalized; performance could be improved further through manual knowledge about specific one task. Cao et al. (2023b) Also, we focus on difficult task where normal method may completely fail, so normal detection dataset like COCO Lin et al. (2014) is not within the scope.

## 4  *DeSSeR*: *De*COMPOSED *S*EGMENTATION WITH *Se*LECTIVE *R*E-LOCALIZING

In this section, we combine the strategy in Section 3 and propose *De*composed *S*egmentation with *Se*lective *R*e-localizing(*DeSSeR*).

## 4.1  METHOD

The method contains (1) decomposition of recognition, localization and segmentation (2) instance-aware prompt to solve counting (3) selective re-localizing with MLLM gridding to improve reliability (4) cost-effective box-point prompt , as shown in Algo 1. Notably, the process contains some hyperparamters, which we show our setting in the bracket: the iou threshold (0.7), the resampling time (1), the gridding layout number (6x6 when images smaller than 384x384, 8x8 when images larger than 384x384) in selective re-localizing.
**Prompt.** The final prompt can be seen in appendix.

**Implementation.** In the main experiment, we use GPT-4o, CogVLM and SAM for recognizer, localizer, and segmentor respectively. For other experiments, the choice of models are the same unless specified. For segmentor using text prompt, we use SEEM Zou et al. (2023) model. All experiment can be done in a single NVIDIA A-6000 GPU.

---

**Algorithm 1** *DeSSeR*

---

1: **Input:** $I$: image, $t$: task, $p$: instance-aware prompts, $n$: times for uncertainty estimate, $\tau$: IOU threshold for uncertainty
2: **Models:** $rcg$: recognizer , $loc$: localizer , $seg$: segmentor
3: **Output:** $M_{pred}$: predicted mask
4: $q \leftarrow rcg(I, t, p)$
5: $box \leftarrow loc(I, q)$
   {Uncertainty Estimation}
6: **for** $i = 1$ to $n$ **do**
7:   $box\_this \leftarrow loc(I, q)$
8:   **if** $IOU(box, box\_this) < \tau$ **then**
9:     $answer \leftarrow rcg(I, \text{``verify } \{box\}, \{box\_this\} \text{ for } \{q\}\text{''})$
10:     {Selective Re-localizing}
11:     **if** $answer ==$ None **then**
12:       $box \leftarrow gridding\_localize(rcg, I, q)$
13:       $box \leftarrow loc(crop(I, box), q)$
14:       **break**
15:     **end if**
16:     **if** $answer == box\_this$ **then**
17:       $box \leftarrow box\_this$
18:     **end if**
19:   **end if**
20: **end for**
   {Prepare cost-effective visual prompt.}
21: $p_{seg} \leftarrow prepare(box)$
22: $M_{pred} \leftarrow seg(I, p_{seg})$

---

## 4.2 TASKS AND DATASETS.

Our experiments are conducted on datasets from three challenging tasks, namely camouflage object detection, zero-shot anomaly detection and polyp segmentation, to showcase our method's broad applicability: (1) CAMO Le et al. (2019), a subset of the CAMO-COCO dataset used in camouflaged object segmentation, includes 250 images for testing in eight categories, featuring diverse challenging scenarios. We used three established metrics: structure-measure $S_\alpha$, weighted F-measure $F_\beta^\omega$, and Mean Absolute Error MAE. (2) CVC300 Vázquez et al. (2017), a benchmark in polyp segmentation in medical images, comprises 60 images of 500x574 resolution from colonoscopy videos. Our evaluations include Mean IOU, Mean Dice, and MAE. (3) VisA Zou et al. (2022), an industrial anomaly detection dataset that contains 12 classes of objects in 3 types (i.e., single instance, multiple instances, and complex structure) with a total 2162 images for testing. For our purposes, we focuses on the 1,200 images in the anomaly subset to evaluate segmentation capabilities. Two widely adopted pixel-based metrics, AUROC and $F1$-max is used.

## 4.3 EXPERIMENT AND DISCUSSION.

We evaluate our proposed *DeSSeR* framework across various challenging segmentation tasks, categorizing the competing methods into specialist, vision general, and LLM general methods, as shown in Table 2. Our training-free approach consistently outperforms existing general and zero-shot/weakly-supervised methods by over 10%, and in some instances, achieves performance comparable to fully supervised models.

Table 4 demonstrates that the quality of semantic expertise significantly impacts segmentation performance, underscoring the importance of leveraging advanced semantic understanding. Furthermore, as illustrated in Table 5, point prompts enhance the prediction quality of LVMs. However, point labels generated by GPT-4V do not lead to performance improvements, indicating potential challenges in fine-grained recognition for LVLMs like GPT-4V.

| Recog. Model | | Accuracy | Loc. Model | | Box IOU | Seg. Model | | Mask MAE |
|---|---|---|---|---|---|---|---|---|
| GPT-4o | LLaVa-1.5 | | CogVLM | LLaVa-1.5 | | SAM | Grounding-DINO | |
| ✓ | | **0.84** | ✓ | | **0.81** | ✓ | | **0.064** |
| | ✓ | 0.53 | ✓ | | 0.55 | ✓ | | 0.110 |
| ✓ | | **0.84** | | ✓ | 0.53 | ✓ | | 0.101 |
| ✓ | | **0.84** | ✓ | | **0.81** | | ✓ | 0.069 |

Table 4: Error analysis on CAMO. This analysis reveals that the prediction errors of our method can be attributed to the model performance at each stage. Moreover, the table shows that selecting appropriate models substantially enhances performancee.

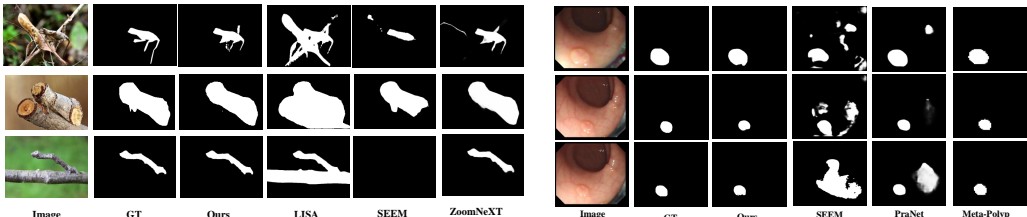

Figure 4: **Qualitative results** of *DeSSeR* on CAMO and CVC300, from left to right respectively.

In the context of anomaly segmentation, Table 6 shows that instance-aware prompts, which involve querying each instance for errors, significantly benefit segmentation performance. Additionally, incorporating overall image descriptions alongside category labels improves localization accuracy.

As presented in Table 6, selective re-localizing is performed frequently and results in performance improvements. Although LLMs can identify inaccurate detections, they may lack the capability to provide accurate results proactively. Comparative experiments between gridding localization by recognizers and box localization from localizers reveal that gridding localization offers robustness when localizers are unreliable, albeit with lower overall accuracy.

Qualitative results in Figures 4 and 5 illustrate that our method produces significantly clearer instance masks compared to other zero-shot methods, particularly in complex scenarios as shown in Figure 5.

| Ways of labeling points | CVC300 | | |
|---|---|---|---|
| | mIOU ↑ | mDice ↑ | MAE ↓ |
| No labels | 0.820 | 0.809 | 0.084 |
| Fixed labels | **0.833** | **0.818** | 0.064 |
| GPT-4o generated | 0.835 | 0.828 | **0.060** |
| Consistent labels | 0.824 | 0.810 | 0.061 |

Table 5: Performance on CVC300 with variant point prompts for segmentation model. The results demonstrates that point prompts enhance the quality of predictions from LVMs. Additionally, point labels generated by GPT-4V do not improve performance, indicating that LVLMs such as GPT-4V may face challenges in fine-grained recognition.

| Prompt for Localizer | VisA | |
|---|---|---|
| | $AUROC_{px}$ | $F1\text{-max}_{px}$ |
| Fixed | 65.6 | 7.1 |
| Category | 80.1 | 24.4 |
| Category + Description | 85.5 | **30.1** |
| Category + Description + Instance-aware Prompt | **86.4** | **32.5** |

Table 6: Performance on VisA with varied prompts for LMM during Localization Stage. The results highlight that incorporating semantic information, such as the target's category, size, and color, significantly enhances the quality of box localization in LMM.

# 5 LITERATURE REVIEW

**Large Vision Models** Recently, SAM Kirillov et al. (2023) proposes a strong universal segmentation model and evaluations show that its zero-shot performance is often competitive with prior results achieved through full supervision. Despite being capable of performing routinary segmentation, the SAM-based models fail when encountering challenging scenes Chen et al. (2023a). Therefore, a zero-shot approach that is good at handling segmentation on scarce and complex data (e.g., COD,

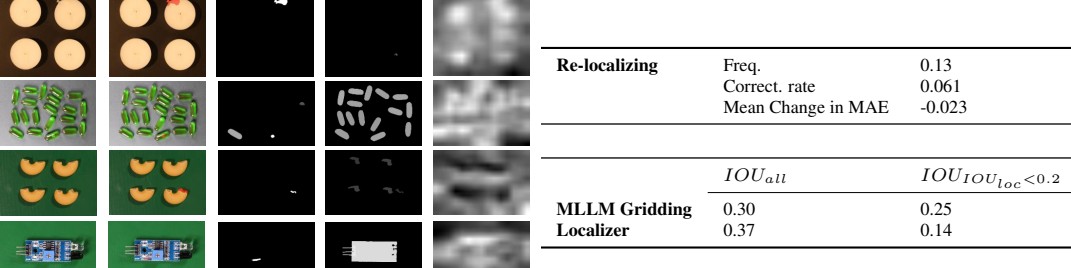

| Re-localizing | Freq. | 0.13 |
| | Correct. rate | 0.061 |
| | Mean Change in MAE | -0.023 |

| | $IOU_{all}$ | $IOU_{IOU_{loc} < 0.2}$ |
| --- | --- | --- |
| **MLLM Gridding** | 0.30 | 0.25 |
| **Localizer** | 0.37 | 0.14 |

Figure 6: Re-localizing and Gridding localization performance in COD dataset.

Figure 5: **Qualitative results** of our method *DeSSeR* on VisA.

AD, and medical image segmentation) is in pressing demand. **Multimodal Large Language Models** Large Multimodal Language Models (MLLMs) Liu et al. (2023b;a); OpenAI (2023a); Zhu et al. (2023); Wang et al. (2023); Ferret (2023); Lai et al. (2023) integrate LLMs with visual capabilities to extend their impressive abilities from language tasks to vision-related tasks. LLaVA Liu et al. (2023b;a), an open-source LVLM, is trained end-to-end by instruction tuning on language-image dataset generated by GPT-4. MiniGPT-4 Zhu et al. (2023) introduces a model combining a frozen visual encoder with the LLM Vicuna Peng et al. (2023). GPT4 Omni (gpt-4o) OpenAI (2023b), an extraordinary model trained by OpenAI, is now the most capable LVLM. However, these LVLMs primarily focus on semantic understanding of images and are not capable of visual grounding. To solve the problem, CogVLM Wang et al. (2023) proposes a LVLM that is able to ground objects to bounding boxes. It uniquely integrates a trainable visual expert module into the frozen pretrained language model and image encoder. **Challenging Segmentation Tasks** (1)Camouflage Object Dectection. In the past years, there has been significant effort in the COD task Fan et al. (2020a); Pang et al. (2023); Chen et al. (2023b); He et al. (2023). ZoomNext Pang et al. (2023) develops a unified collaborative pyramid network which leverages multi-head scale integration and rich granularity perception units. WSCOD He et al. (2023) introduces the first weakly-supervised camouflaged object detection method using scribble annotations, featuring a novel consistency loss, a feature-guided loss, and a new network. CPVLF Tang et al. (2023) introduces the camo-perceptive vision-language framework to assess if MLLM can adapt to COD without specific training. Its still have large gaps with fully-supervised ones. (2)**Zero-Shot Anomaly Dectection** Anomaly detection has attracted great interest in various domains, e.g., industrial quality control Bergmann et al. (2019; 2020); Zou et al. (2022) and medical diagnoses Baur et al. (2021). Recently, zero-shot anomaly detection (ZSAD) is proposed as a promising setting where neither normal nor abnormal image is provided. (3)**Polyp Segmentation in Medical Images** The pioneering work Fan et al. (2020b) presents a novel method, the Parallel Reverse Attention Network (PraNet) for accurate polyp segmentation; Polyp-PVT Dong et al. (2021) proposes a polyp segmentation model using a transformer encoder and three innovative modules for robust performance on five polyp datasets. More recently, Polyp-SAM++ Biswas (2023), a variant of SAM, utilizes text prompts for polyp segmentation, whose evaluation shows its effectiveness compared to un-prompted SAM, but still have large gaps on some datasets (e.g., CVC300) compared to SOTA methods.

## 6 CONCLUSION

In this paper, we introduce *DeSSeR*, a novel and training-free strategy leveraging Large Multimodal Language Models (MLLMs) for segmentation in challenging tasks. Our study shows the pivotal role of semantic understanding in such tasks, a facet where general large vision models (LVMs) often fall short. The method uniquely combines the semantic insights from MLLMs with the visual distinction capabilities of LVMs. Abundant experiment and analysis are done to provide insights on the challenges faced when approaching zero-shot segmentation with MLLMs, and the logic behind the *DeSSeR* method. Our comprehensive experimental analysis across various datasets not only showcases the efficacy of *DeSSeR* but also suggests its genration on various tasks and datasets.

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
