# OpenReview forum: "Seeing is Knowing: Advancing Semantic Understanding with MLLMs in Grounding Tasks"
_ICLR.cc/2025/Conference — ICLR 2025 Conference Withdrawn Submission_

### Official Review · Reviewer_hznL · 2024-10-20

**Soundness:** 2
**Presentation:** 1
**Contribution:** 1
**Rating:** 3
**Confidence:** 3

**Summary:**

This paper aims to enhance the performance of large vision models in grounding tasks by leveraging the semantic understanding capabilities of Multimodal Large Language Models (MLLMs). This paper introduces Decomposed Segmentation with Selective Re-localization (DeSSeR), which decomposes the segmentation flow into recognition, localization, and segmentation steps. The proposed DeSSeR demonstrates strong zero-shot performance in camouflage object detection, zero-shot anomaly detection, and polyp segmentation.

**Strengths:**

- The proposed DeSSeR achieves strong zero-shot performance across diverse tasks, including camouflage object detection, zero-shot anomaly detection, and polyp segmentation.

**Weaknesses:**

- There are many typos that affect readability and clarity.
- DeSSeR mainly builds on sequential combinations of existing models such as MLLMs and vision foundation models, providing limited technical novelty for the ICLR community.
- I think that the performance of DeSSeR is highly dependent on utilized models (e.g., GPT-4o, CogVLM, and SAM), raising concerns about its independent contribution.
- Although the authors propose techniques to improve grounding multiple instances and target misses,  experimental results to validate their effectiveness is not provieded.
- It is unclear how the proposed DeSSeR performs on general visual grounding datasets (e.g., [1], [2]).
- Performance evaluation based on the hyper-parameters in Algorithm 1 is required for a more comprehensive analysis.

[1] Rasheed, Hanoona, et al. "Glamm: Pixel grounding large multimodal model." Proceedings of the IEEE/CVF Conference on Computer Vision and Pattern Recognition. 2024.
[2] Yuan, Yuqian, et al. "Osprey: Pixel understanding with visual instruction tuning." Proceedings of the IEEE/CVF Conference on Computer Vision and Pattern Recognition. 2024.

**Questions:**

- What is the technical novelty of the proposed DeSSeR compared to existing methods?
- Could you provide additional experimental results demonstrating the performance of DeSSeR on general visual grounding datasets (e.g., GLAMM, Osprey)?
- How sensitive is the performance of DeSSeR to the hyper-parameters in Algorithm 1?

(Please see the Weakness)

---

### Official Review · Reviewer_ktkt · 2024-10-26

**Soundness:** 2
**Presentation:** 1
**Contribution:** 2
**Rating:** 3
**Confidence:** 4

**Summary:**

This paper proposes a new framework that utilizes MLLMs to assist VLMs in handling challenging segmentation tasks. The framework consists of a recognizer, followed by a localizer and a segmentor, performing segmentation in a training-free manner. Results on camouflage object detection, zero-shot anomaly detection, and polyp segmentation demonstrate the effectiveness of the proposed method.

**Strengths:**

1. A new training-free method is proposed to perform segmentation in some chanllenging scenarios.
2. Some interesting analyses are provided to showcase some abilities and properties of existing MLLMs and LVMs.
3. Experiments on several tasks are conducted to verify the effectiveness of the proposed method.

**Weaknesses:**

1. The writing quality of this paper is poor. There are too many typos, which severely affect the reading experience. The authors need to carefully revise the paper to improve the writing quality.
2. This paper combines MLLMs (such as GPT-4) and an LVM segmenter (such as SAM). Many previous works, such as GSVA [a], have also used both. The authors need to show the performance of these methods on the evaluated datasets and provide a comparison. However, the authors only provide the results of LISA, which is insufficient.
3. In Section 3.2, the authors state that the paper aims to address two issues: grounding multiple instances and target misses. However, no experimental results are provided to demonstrate that the proposed method has a significant advantage over previous approaches in addressing these issues. Although Table 6 provides results on the VisA dataset, not all images in VisA contain multiple targets, so these results cannot sufficiently demonstrate the proposed method’s advantage in multi-instance grounding.
4. More ablation studies on the hyperparameters used in this method, such as the IoU threshold and the gridding layout number, should be conducted to evaluate the robustness of the hyperparameter settings.
5. The authors state that the final prompt can be found in the appendix, but no appendix is provided.


[a] Gsva: Generalized segmentation via multimodal large language models, CVPR2024

**Questions:**

Please see the weakness section.

---

### Official Review · Reviewer_ie2t · 2024-10-30

**Soundness:** 3
**Presentation:** 3
**Contribution:** 2
**Rating:** 5
**Confidence:** 4

**Summary:**

Through preliminary exploratory experiments, the authors observe that while large vision models (VLMs) have achieved substantial success in most everyday scenarios, they encounter difficulties with specialized grounding tasks. This limitation stems primarily from the current vision models’ limited semantic understanding of both tasks and images. In contrast, large multimodal language models (MLLMs) excel in semantic comprehension and instruction-following but lack precision in detailed recognition. To address this gap, the authors propose utilizing MLLMs to support VLMs in tackling challenging zero-shot segmentation tasks. Experimental results demonstrate notable performance improvements in complex segmentation tasks, including camouflage object detection, anomaly detection, and medical image segmentation, compared to standard zero-shot baselines.

**Strengths:**

The overall logical flow of this paper is very clear, progressing from a well-defined research motivation to a structured problem-solving approach, which effectively guides the reader through the authors' research rationale. Furthermore, the comparative experiments are relatively thorough, with an abundance of ablation studies and visualizations that showcase the effectiveness of the proposed architecture. Additionally, the authors provide a dedicated discussion of the paper’s "Scope and Limitations", which adds depth to the evaluation and contextualization of their findings.

**Weaknesses:**

1. The manuscript uses the ICLR 2024 conference template, which should be updated to the 2025 version. The authors are advised to make this correction.
2. While the research motivation is clear and straightforward, the approach of leveraging MLLMs to enhance zero-shot segmentation performance is not a novel topic. Furthermore, the authors' method relies on directly modular combination of different models to achieve improved results, which limits the architectural innovation in the proposed design.
3. The experimental comparisons lack several important baselines. For example, in Table 2, the models categorized as LVLMs with segmentation abilities are limited to LISA, while other relevant models, such as "GLaMM：Pixel Grounding Large Multimodal Model", are not included. Additionally, recent foundational models like SAM 2 should also be included in Tables 2 and 3 for a more comprehensive quantitative comparison.
4. Some figures, such as Figure 3, could benefit from adjustments of font size to enhance readability and visual appeal.

**Questions:**

1. Respectfully, I suggest that the authors reclaim the innovation of their proposed DeSSeR architecture. While I acknowledge its effectiveness, the novelty of such a concatenated structural design remains unclear.
2. The authors should update the paper to the latest conference template.
3. To thoroughly validate the advantages of the proposed method, more baseline comparisons should be included.
4. If time allows, additional experiments on a broader range of tasks or datasets could further demonstrate the effectiveness and generalizability of the proposed method and its insights.

---

### Note · Authors · 2024-11-17

I have read and agree with the venue's withdrawal policy on behalf of myself and my co-authors.